# Does Academic Publishing Lead to Work-Related Stress or Happiness?

**Jaroslava Kubátová** 

Faculty of Arts, Department of Applied Economics, Palacký University Olomouc, Křížkovského 12,
771 80 Olomouc, Czech Republic; jaroslava.kubatova@upol.cz

**Abstract:** The topic of work-related stress and happiness has recently been of interest to science as well as in practice. Work-related stress has negative effects on workers, organizations, and the whole of society, whereas happiness has positive effects. It is therefore important to monitor the wellbeing of workers. This article deals with stress and happiness as related to academic publishing. To answer the research question of whether academic publishing leads to stress or happiness, a narrative analysis was conducted. Narratives from ten Czech academics were collected and analyzed with the use of a categorical-content approach. The categories used are the general causes of work-related stress and happiness as identified in the literature: work overload, ambiguity, conflict, the sense of meaningful work, job satisfaction, and affective organizational commitment. It was found that academic publishing leads to both work-related stress and happiness. However, stress is more prevalent. Not only do academics experience all the general causes of work-related stress, unfortunately they often lack the sources of happiness. Many specific causes of stress and happiness, as well as unhappiness, were discovered in the narratives. Several ways to improve the situation have been suggested. Refining policies in human resources is particularly important if universities wish to retain their academics.

**Keywords:** academic publishing; Czech Republic; work-related stress; happiness at work

## 1. Introduction

Work-related stress has been of scientific interest since the 1950s [1] and research in this area has continued to grow [2]. With the development of positive psychology in the late 1990s, there has also been a growing interest in work-related happiness, often called happiness at work [3]. The topics of work-related stress and happiness also apply to academic environments [4]. An important part of academic work around the world is publishing. Academic publishing in high-quality journals determines the ranking of an institution. Based on their publications, academics are evaluated and potentially awarded further research grants. The whole realm of academic publishing is continually evolving and it raises many questions. Today, the advanced digital technologies provide greater integration of global research communities than at any time in the past. An emergent global ecosystem of scholarly communications, still largely dominated by Anglo-America, is ushering in an era that enables us to talk of 'new knowledge ecologies' and 'three ages of the journal' as the academic world moves from text to electronic and video communication. The changes in the global knowledge ecosystem emphasize new concerns for the geographical distribution of journal knowledge and also the effects of global altmetric and peer review systems on scholarly life [5] (p. 1402). This article focuses on the question of whether academic publishing, as one of the expected outputs of academic work, leads to stress or happiness in academics.

To answer this question, it is necessary to first explain the theoretical background. Based on the managerial and psychological literature, the concepts of work-related stress and happiness are

introduced, and their importance is explained. Later, the method of narrative analysis is applied. With the use of a categorical-content approach, the causes of stress and happiness, as relating to academic publishing, are identified. Potential ways to improve the situation are also discussed.

## 2. Work-Related Stress

Work-related stress is popularly described as occurring when there are discrepancies between the physiological demands within a workplace and the ability of employees to either manage or cope with such work demands [6]. Work-related stress is associated with sizeable financial costs for organizations and national economies [7]. In general, the costs are associated with absenteeism and presenteeism, loss of productivity, health care costs, and disability benefit payments. Eliminating these costs is just one of many reasons to reduce work-related stress as much as possible.

Trenberth and Dewe [8] argued that it is difficult to find a unitary definition of work-related stress. Michie [9] concluded that stress is related to the interaction between the situation and the individual. Stress is therefore more likely in certain situations and in specific individuals than in others. Selye [10] (p. 15) suggested the following definition of stress: "Stress is the nonspecific response of the body to any demand." All endogenous or exogenous agents that make such demands are called stressors. Selye pointed out that it is necessary to distinguish between two types of stress effects, eustress (from the Greek eu - good) and distress (from the Latin dis - bad). Depending on conditions, stress is associated with either desirable or undesirable effects. Selye argues that there is no specific stress; however, it is acceptable to typify stress if it is clear that it refers to the stress produced by specific factors. In this article, based on Selye's theory, work-related stress means distress caused by work-related factors which are further called stressors.

The late 1970s produced work that extended the understanding of the range of work stressors [11]. Since then, several models of work-related stress that include specific stressors have been suggested [12–18]. Altogether, more than 80 stressors have been specified in these models and theories. Lazarus and Folkman [14] (p. 238) argued that the critical factors in creating stress are conflict, ambiguity, and overload. Conflict arises when meeting a demand violates a strongly held value or exceeds the role of the person. Ambiguity occurs when the person is unclear as to what is expected. Overload occurs when the demands or requirements exceed the person's resources. After analyzing the specific stressors suggested in the models and theories mentioned above, we can find that each of them can be classified as one or more of the three general critical work-related stressors identified by Lazarus and Folkman, i.e., overload, ambiguity, or conflict.

In academia, management by performance indicators, such as the number and quality of scientific publications, and performance-based funding mechanisms were found to be sources of stress [19]. Unfortunately, the pressure to publish not only heightens stress levels but it also has other negative effects, such as marginalizing teaching, or promoting research that may lack relevance, creativity, or innovation [20]. Publishing in English—considered a norm nowadays—was found to be particularly challenging for non-native speakers [21].

## 3. Happiness at Work

Even though the term happiness is frequently used, it is also not unambiguously defined [22]. The same applies for the term 'happiness at work'. For example, Lyubomirsky et al. [23] (p. 820) did not define happiness at work but they identified happy individuals as those who experience high average levels of positive affect. Fisher [24] (p. 385) concluded that all constructs of happiness at work refer to pleasant judgements (positive attitudes) or pleasant experiences (positive feelings, moods, emotions, flow states) at work.

Price-Jones [25] (p. 4) defined happiness at work as " . . . a mindset which allows you to maximize performance and achieve your potential. You do this by being mindful of the highs and lows when working alone or with others." Simon-Thomas [26] described happiness at work as feeling an overall sense of enjoyment at work; being able to gracefully handle setbacks; connecting amicably with

colleagues, coworkers, clients, and customers; and knowing that your work matters to yourself, your organization, and beyond.

Johnson, Robertson and Cooper [27] argued that happiness is quite similar to psychological well-being (PWB). A key factor in PWB is work that is rewarding, involving good relationships with colleagues, and opportunities to feel a sense of achievement on a regular basis. Workers with high PWB feel a sense of purpose and positive emotions (they feel good at work). Johnson, Robertson, and Cooper pointed out that PWB is linked with a very wide range of important outcomes for organizations, such as high productivity and lower absenteeism, as well as presenteeism. They suggested that the main workplace factors of PWB are as follows:

- control and autonomy, i.e., the extent to which workers feel that they have control over how they carry out their work,
- work (over)load, i.e., the extent to which the workload itself is a source of excessive pressure for an individual due to e.g., unrealistic deadlines or unmanageable amount of work,
- resources and communication, i.e., the extent to which workers have the necessary resources to perform their job and to which the communication at the workplace is effective,
- job security and change, i.e., the extent to which workers consider their jobs as stable or to which they anticipate some changes,
- work relationships, i.e., the extent to which workers consider the relationships with other people good,
- job conditions, i.e., the broad context within which workers are expected to perform their jobs.

Sirota and Klein [28] argued that employees seek to satisfy three key needs: equity, achievement, and camaraderie. When these needs are met, workers are happy and ready to enthusiastically accomplish organizational goals. More exactly, it means that employees need to feel that they are respected, treated, and compensated in a fair way. They should feel that their jobs are secure, be sure they have the resources to perform their jobs, and that there are no unnecessary obstacles such as excessive bureaucracy. The job should also be reasonably challenging. Feedback, recognition, and rewards should be provided. Partnership and teamwork should be supported and developed. The role of effective leadership is critical. These findings are in accordance with Fisher's [24] conclusions, but in addition, she suggested adopting high-performance work practices within the organization.

Steger [29] pointed out that happiness at work is tied to the sense of meaningful work and that workers who see meaning in their work are more committed, engaged, and productive. Salas-Vallina and Alegre [30] argued that happiness at work can be simply defined on the basis of three dimensions: engagement, job satisfaction, and affective organizational commitment. They defined engagement as a "special feeling of energy and motivation related to the capacity to feel thrilled, vibrant, excited or passionate at work" (p. 4) and deduced that engagement refers to the feelings that result from meaningfulness at work. Job satisfaction refers to the judgements of job characteristics. Unlike engagement, which is related to the mood of the worker, job satisfaction refers to feelings about salary, career opportunities, relationships with coworkers, and other working conditions. Affective organizational commitment refers to emotional attachment, identification, and involvement in the organization as a whole.

In this article, based on Fisher [24], happiness at work is understood as pleasant judgements and pleasant experiences (positive feelings, moods, emotions) of workers. Happy workers are more likely to show superior performance and productivity and to handle managerial jobs well. On the other hand, they are less likely to show counterproductive behavior, absenteeism, turnover and job burnout [23]. Happy workers take considerably less sick leave, are more energized, more productive, and intend to stay in their organizations [31]. These arguments are just a few among many to support happiness at work. Yet, not much is known about the sources of happiness in academic work. Miller et al. [20] argued that the high social reputation of academic staff in society and academic autonomy are the sources of their satisfaction.

## 4. Method

To answer the question of whether academic publishing leads to work-related stress or happiness, a narrative analysis was conducted. Narratives are tools to explore aspects of human experience. When narrators tell a story, they give a narrative form of their experience. Narratives attempt to explain or normalize what has occurred; they lay out why things are the way they are or have become the way they are. A narrative provides a portal into the realm of experience, where speakers lay out how they, as individuals, experience certain events and confer their subjective meaning onto these experiences [32].

The narrative interview envisages a setting that encourages and stimulates interviewees to tell a story (narrative) about a significant event in their life and its social context [33]. The purpose of the narrative interview is to provide an opportunity for the participant to narrate his or her experience for the researcher. The technique of narrative interview does not impose strict discourse guidelines on the interviewees. The researcher asks an open-ended question that invites the interviewees to respond in narrative forms and the interviewees are encouraged to be the ones to decide how and what to tell (narrate) [34]. Using a narrative analysis, the researcher then interprets the collected stories.

In this research, a categorical-content approach, also known as content analysis [35], is applied. Categories of the studied topic are defined, and separate utterances of the narratives are extracted, classified, and sorted into these categories. In accordance with the research aims and questions, the contents collected in each category are used descriptively to formulate a picture of the studied topic.

In this research, the categories are the general stressors and causes of happiness at work. The general stressors are work overload, ambiguity, and conflict. Even though they were identified several decades ago [14] they still occur in the modern workplace [18] and they are particularly prevalent in the environment of knowledge work [36]. Work overload is the situation in which people have the feeling that they have too much work to do. It is related to the difficulty, number, or complexity of the job tasks. Ambiguity is the situation in which people are not sure what are they supposed to do or how they are supposed to work. Conflict reflects either difficult interpersonal relationships at the workplace or there may be also a conflict of roles. This can be a conflict of work roles and/or personal roles.

The dimensions suggested by Salas-Vallina and Alegre [30] and defined above with one adjustment are suggested as the general causes of happiness at work. Instead of engagement, the sense of meaningful work is applied as one of the categories. Salas-Vallina and Alegre argued that engagement stems from the feeling of meaningful work. It is also in accordance with Steger's [29] findings or with Walczak and Derbis [37], who argued that seeing work as meaningful is a way to happiness.

Therefore, the categories for this narrative analysis are as follows: work overload, ambiguity, conflict, the sense of meaningful work, job satisfaction, and affective organizational commitment. In the collected narratives, the sections that relate to these categories were identified and analyzed to answer the research question.

## 5. Collected Narratives and Their Categorical-Content Analysis

Narratives from 10 Czech academics were collected. To gather their short thematic stories, interviews were conducted in the manner of conversations in everyday life [38]. The introductory question was: Academic publishing is a part of academic work. What does academic publishing mean for you? The whole research was designed and conducted just by the author. The narratives were collected between April and June 2019. The conversations were conducted face to face in Czech in a public space (a café or restaurant) and on average, they lasted five minutes. There has been some cooperation between the participating academics and the author in the past. Based on already established common trust, the participants were asked to discuss the topic of academic publishing for a research project. However, they were not informed that the research focuses on stress and happiness related to academic publishing. With the consent of the participants, the conversations were recorded and without delay transcribed. Later, the transcriptions were translated into English.

The narrators were academics from four Czech public universities. All of them have experience with publishing in English. Recently, publishing in English has been considered standard in the Czech Republic, even for Ph.D. students. All participants also have professional experience from an English-speaking environment. Further details about their fields, age, and years of tenure are provided in the list:

- Robert, Ph.D., Professor, IT, aged 54, 23 years in academia, University 1
- Talia, Ph.D., Assistant Professor, Philology, aged 36, 11 years in academia, University 1
- Anna, Ph.D., Associate Professor, Management, aged 55, 21 years in academia, University 2
- Dan, Ph.D., Associate Professor, Linguistics, aged 46, 19 years in academia, University 2
- Kathy, Ph.D., Assistant Professor, Psychology, aged 37, 13 years in academia, University 2
- Jake, Ph.D., Assistant Professor, Mathematics, aged 34, 9 years in academia, University 2
- Oliver, lecturer and Ph.D. student, Finance, aged 29, 3 years in academia, University 2
- Tobin, lecturer and Ph.D. student, Statistics, aged 33, 8 years in academia, University 2
- Lucia, Ph.D., Associate Professor, Management, aged 56, 24 years in academia, University 3
- Allan, Ph.D., Assistant Professor, Soc. Anthropology, aged 34, 8 years in academia, University 4

To keep the narrators anonymous, their names were changed, and their universities are not specified. Professor is the highest academic position in the Czech Republic, followed by, in descending order, associate professor, assistant professor, and lecturer. Six narrators were men, four were women, one was a professor, three were associate professors, four were assistant professors and two were lecturers. The lecturers were the only ones to have completed a Ph.D. program but both were approaching their final exams and doctoral thesis defenses.

The transcripts of the most relevant parts of the narratives are presented below. When the narrator mentions a stressor or cause of happiness, it is identified (coded) in parenthesis. The categories are marked as follows:

- WOV—Work overload
- AMB—Ambiguity
- CON—Conflict
- SMW pos, SMW neg—The sense of meaningful work (pos means positive, i.e. the cause is present, neg means negative, the absence of the cause is mentioned by the narrator)
- JSA pos, JSA neg—Job satisfaction (ditto)
- AOC pos, AOC neg—Affective organizational commitment (ditto)

Robert, Ph.D., Professor, University 1:

*Well, it's important to publish, to build your reputation, as well as the reputation of your university. (AOC pos) But the ever-changing rules in this country exhaust me. (AMB) And there are so many other things to do. I've got several Ph.D. students and it's a lot of work to supervise them. (WOV) And of course we publish things together. But during the last assessment I was criticized that I used my Ph.D. students to have more publications and they recommended to publish more on my own. But to me that doesn't make any sense. (JSA neg)*

Talia, Ph.D., Assistant Professor, University 1:

*It's crazy, the requirements for scientific work and our publications are constantly changing. (AMB) It used to be enough to publish in conference proceedings but not anymore. It used to be considered a success to publish a book. Now when you publish a book, of course it must be in English, and it must be in a "prestigious publishing house". But no one tells you what a "prestigious publishing house" is. (AMB) Then to fulfill the job requirements we're supposed to publish in journals that are indexed in ERIH, Scopus or Web of Science. I've been able to publish several articles in various journals in ERIH. But about a year ago ERIH was excluded, so now only Scopus and Web of Science count.*

*To make things worse we are 'recommended' to publish in high-ranking journals. But the competition is so fierce there. (JSA neg) And you know, we're not actually scientists, we're university professors. (CON) Scientists don't have to teach, or maybe they do but not that often. And they have time to conduct proper research and to prepare their articles. And of course, they get the money for their research. I am so confused and exhausted by this all. (WOV)*

*And it is so hard to write an article and teach at the same time. (WOV) But because we have to publish then the teaching ends up being somehow shortchanged. (CON) I mean I would love to spend more time preparing my lectures but too often I simply cannot find the time. (JSA neg)*

Anna, Ph.D., Associate Professor, University 2:

*The work pressure is tremendous. (WOV) My boss told me that an academic of my rank is supposed to have two articles in Web of Science or Scopus journals every year. In general, I agree. But regarding my other roles at the department, I have so much paperwork that during the teaching and exam periods I'm not able to find the time to concentrate on another publication. (CON, JSA neg) The result is that I end up spending my vacations trying to write something. I just don't have enough time to have a real break and of course I feel guilty that I don't spend enough time with my family. (CON)*

*I think it's fine if you're able to publish something interesting no matter if it's a book or an article. It could have an impact somewhere. You can push development in your field even if it's just little bit. And you can use them as reading for your students. Then you don't need to look for other readings and the students can see you're an expert because you've published something. (SMW pos) I remember seeing my first publication in the library catalogue. It was a book. And I thought "Wow, I'm actually immortal now. This record will be here long after I'm gone." (SMW pos)*

Dan, Ph.D., Associate Professor, University 2:

*Our University just tells us to publish. We're told that it's in our own interest to publish or we'll lose our accreditations which means we'll lose our jobs. They don't care how busy we are or if we have the time or the resources to do some research. They just want results. (AOC neg) At the end of the year if there's any money left in the budget you might get a bonus for your publications but then you might not. (AMB, JSA neg)*

*Recently, an article I wrote with a colleague was accepted in a journal that's indexed at the Web of Science and I'm happy with it. (JSA pos) But the reviewers asked us for revisions. They called it 'small' revisions. But in fact the revisions changed what we wanted to communicate quite a lot. Of course, in theory you can explain and defend your point of view, but then you risk they will reject you. And you need to be published because you have to report your results. (CON) So you better agree to do the revisions. (SMW neg)*

*I like to publish once in a while when I really have something new to share or discuss with the community. (SMW pos) But because we are evaluated according to the numbers and quality of our publications, which as we know is not defined, (AMB) I find myself producing publications just for the sake of publishing. (SMW neg)*

Kathy, Ph.D., Assistant Professor, University 2:

*I have my little son now and we're planning another child. So I had to tell my boss that I'm not able to publish in quantity and quality he requires. I do like the teaching, and I can still follow developments in my field but I cannot publish. (CON) And my boss was fine with that, but in reality I'm not. (JSA neg) I feel like I should be doing everything my colleagues are supposed to be doing. (CON) Of course this influences the interpersonal relations in our department. Even though I negotiated my own working conditions I'm still not happy. It seems impossible to be a good mother and a successful academic at the same time. (CON)*

*When I eventually submit an article, it's so stressful to wait for the email from the journal (JSA neg) and then to learn your article was rejected. (JSA neg) I know it's normal, but last time I felt like crying. It took me so much time to write the article and so far it's been for nothing. (SMW neg)*

Jake, Ph.D., Assistant Professor, University 2:

*I try to get published in the highest-ranked journals to support my department. (AOC pos) I want to show that we can do this! But it might take a year and a half to learn the result which is really awful. (JSA neg) If the article is accepted then that's great (JSA pos) but still, the time between writing and publication is too long. And the university wants you to publish a certain number of articles in the good journals every year, but it's impossible to plan the publications. (JSA neg) And if it's not accepted it can be really depressing. (SMW neg)*

Oliver, lecturer and Ph.D. student, University 2:

*A lot of people here are complaining about publishing but I've been quite lucky so far. My research topic is quite new in our region. So I actually don't have much trouble to publish my work. (JSA pos) And I'm happy to share my results. It seems like my research has meaning. (SMW pos)*

Tobin, lecturer and Ph.D. student, University 2:

*I've submitted four articles with my colleagues in the last eighteen months. But we don't know the results yet. (AMB) So we've got nothing to report to the university and it looks like we haven't written anything, even though we've been working very hard. (CON, JSA neg)*

*And money for research is a vicious circle. To do research and to publish you need some money. But to get money from the faculty or from a grant you're supposed to have some publications of a certain quality. (JSA neg) It's not uncommon that I have to spend my own money on books, or on proofreading for example. But you better not tell this to my partner! (CON)*

Lucia, Ph.D., Associate Professor, University 3:

*I once discussed all these problems with publishing with a colleague of mine in front of her daughter. She worked for an analytics company, for a better salary then we had. And she just asked "Why on earth are you doing this?" I also ask myself this question, why? (SMW neg)*

*It's particularly difficult to compete with native speakers. It's so much easier to write in your mother tongue. (JSA neg) But even the native speakers are not sure what is correct. I had an article proofread by a native speaker who was also an expert in the field. But the reviewer asked to improve the English in the article. Based on the advice by my proofreader, I ended up having a big argument with the reviewer about one "the" in my text … (CON) It drained me of so much energy. (WOV) Sometimes you get rejected so quickly, in a matter of hours or overnight, they probably didn't bother to read the article. (SMW neg) I think they just saw that someone from the old eastern bloc submitted the article and immediately just wrote "not for us". (JSA neg)*

Allan, Ph.D. Assistant Professor, University 4:

*With all these publications, teaching, paperwork, exams, reviews, supervisions and so on I feel really overworked. (WOV) Sometimes I feel like leaving academia and finding a job where you can keep your head clear. (JSA neg) But then I realize that at least we've got a lot of flexibility. I don't need to publish daily from 9 to 5. Academic freedom and flexibility are the benefits of academic work. (JSA pos)*

*And I enjoy the networking which comes with publishing. You don't need to go to a conference nowadays to meet people you can communicate or cooperate with. I'm on Researchgate, on Google Scholar, I'm a member of a few expert groups on Facebook and I've met so many interesting people there. I like sharing knowledge with them. (JSA pos)*

## 6. Results and Discussion

To answer the research question, whether academic publishing leads to work-related stress or happiness, a categorical-content analysis of the collected narratives was carried out. In Table 1, the number of identified occurrences (in parenthesis in the first column) and its causes (in the second column) are identified for each category.

**Table 1.** The causes of stress and happiness related to academic publishing.

| **Work-Related Stress** | |
| --- | --- |
| Work overload (WOV, 6) | A lot of work, many things to do, exhaustion, work pressure, much needed energy, overwork |
| Ambiguity (AMB, 6) | Changing rules, changing requirements, missing criteria, long wait for the decision about publications acceptance or rejection |
| Conflict (CON, 11) | Scientist vs. professor, teaching vs. publishing, work vs. family, working vs. resting, one's own opinion vs. requirements of the reviewers, parent vs. teacher vs. scientist, conflicts with colleagues, hard work vs. no reportable results, one's own money used for research and publication, quarrel about use of English |
| **Happiness at Work** | |
| The sense of meaningful work (SMW pos, 4; SMW neg, 6) | Positive: Publications as reading for students, proof of expertise, publication lasts forever, knowledge sharing<br>Negative: Changing text to accommodate the reviewers in order to be published, publishing in order to meet the requirements of the university, article rejection, missing meaning of publishing |
| Job satisfaction (JSA pos, 5; JSA neg, 14) | Positive: Being published, flexible work and academic freedom, meeting interesting people online on research platforms or in groups<br>Negative: Unfair criticism by the faculty management, too demanding work requirements, too many different job tasks, too little time to accomplish the job tasks, unclear remuneration policy, unequal working conditions, undefined time limits for acceptance/rejection of submitted publications, too much time to wait for the result (acceptance/rejection), rejection of submitted article, difficulties with work planning, poorly defined criteria for academic performance appraisal, conflicting criteria for academic performance appraisal, difficulties with publishing in English (as a foreign language), perceived disadvantages of nationality, no boundaries between work and life |
| Affective organizational commitment (AOC pos 2, AOC neg 1) | Positive: Publishing as a way to build the reputation of the university, publishing in order to support one's own department<br>Negative: "Stick" motivation policy to publish used by the university |

Causes of both stress and happiness were mentioned in the narratives. Work overload was mentioned frequently. The academics often complained about the quantity of work and the range of tasks. They regularly work under time pressure. They experience a lot of ambiguity. This ambiguity has three main sources. The first is waiting for the decision on the acceptance or rejection of their publications that can take an unpredictable amount of time. The second is the problem with frequently changing rules for evaluating research organizations at the national level. The third is related to the changing requirements for the number and type of publications that academic staff are expected to produce and a lack of evaluation criteria at the university level.

Many different conflicts are related to academic publishing: conflicts of work roles, for example whether they are more like scientists or professors (i.e., teachers); conflicts of work activities relating to dividing time between lecture preparation and writing; conflicts between work roles and life roles, such as parenting and academic duties; conflicts in their work-life balance in general. The academics

also reported interpersonal conflicts with their colleagues and with their reviewers. The necessity to accommodate the requirements of the reviewers was presented as an inner conflict. The necessity to obtain money for research and publication was also mentioned as leading to inner conflict. Particularly the conflicts of work roles and work activities are linked with the problem of work overload.

When analyzing the categories of happiness in the narratives, it was found that the academics not only mentioned some positive facts, but they also very clearly specified the missing factors of happiness. When these specific factors were assigned to the respective categories, they were marked either as positive, i.e., increasing the happiness, or negative, i.e., decreasing the happiness or even causing unhappiness. Overall, more negative than positive factors of happiness were found in the narratives.

Publications were seen as meaningful because they are proof of expertise, they last longer than an academic career, and they support knowledge sharing with other scholars and with students. On the other hand, when an article is not accepted for publication, or it has to be changed to follow the recommendations of the reviewers, the work is perceived as meaningless. This feeling occurs although the academics can realize that the reviewers just intended to improve the papers by providing objective feedback. The sense of meaninglessness was also related to excessive technical requirements of the university and to the missing purpose of publishing in general.

Among all six categories, job satisfaction was the most frequently mentioned, however, more often in the negative than in the positive sense. The acceptance of a publication, the flexibility and freedom of academic work, and the presence and cooperation in virtual (online) research communities contributed to an increase in job satisfaction. On the other hand, many factors that decreased job satisfaction were found in the narratives. Some of them are obviously linked with the identified stressors but here, they are clearly articulated (too many different tasks, too demanding requirements, too little time, problems with acceptance/rejection of publications, problems with work-life balance). The academics also mentioned a perceived discrimination on the basis of their nationality or native language. They mentioned problems such as poorly defined or conflicting criteria for academic performance appraisal and unfair criticism, unequal working conditions, or unclear remuneration policy. These indicate that there is room for improvement in the management of academic staff.

Affective organizational commitment was rarely identified in the narratives. One academic mentioned that he saw publishing as a contribution to the university reputation, whereas another wants to support only his department. In one narrative, negative feelings towards the university were apparent. The reason was that the university was threatening the academic staff to motivate them to publish.

The categorical-content analysis of the collected narratives represents the perceived reality of academic publishing. It reveals that publishing causes quite a lot of stress but can also lead to happiness. The academics experience the same general causes of stress and happiness that were identified in other professions [12–18,28–30]. Nevertheless, Simon-Thomas and Newman [39] argued that education staff sees a relatively high purpose in their work and feels engaged, which are prerequisites of happiness at work.

This research has two main limitations. First, the number of the interviewees is limited; secondly, they are all from one country and they are all from universities. Limited numbers of participants are typical for narrative analyses. Despite this limitation, the picture of the examined reality is rich and can be used as a starting point for further research, as well as for proposals of practical changes. The Czech Republic applies the same criteria for academic performance as many other countries around the world. It is therefore reasonable to presume that the causes of stress and happiness related to academic publishing at universities are similar across these countries. On the other hand, Czech culture, as well as Czech academia, are specific. To find out if the presented results also have general relevance outside the Czech Republic, this study should be replicated in other countries.

## 7. Conclusions

In this article, the question of whether academic publishing leads to work-related stress or happiness was addressed. The general answer is that academic publishing leads to both stress and happiness. On the basis of the conducted narrative analysis, it can be concluded that stress is prevalent, but academic publishing can also lead to happiness. The academics experienced work overload, ambiguity, and different types of conflicts. On the other hand, successful publishing is seen as meaningful and satisfying. However, it relates very weakly to affective organizational commitment. Based on the findings of this research, changes at the national level and at the university level can be suggested. In the countries where universities are supervised by a national authority, the standards for academic performance should be clear and valid in the long term. Frequent changes that happen in many countries, including the Czech Republic, are contrary to the long-term nature of academic publishing. Universities must adapt to these changes, but they fail to establish adequately motivating human resource policies. Good job descriptions would help to avoid work overload and many conflicts. Clear and long-term valid rules and criteria for academic performance would decrease ambiguity. The missing causes of happiness at work also give good clues as to what can be improved at universities. If the pressure to publish was not so great, academic staff would not change their texts just to avoid rejection. Moreover, they would see much more sense in publishing. If universities had clear remuneration policies, job satisfaction in academic staff would be higher. Universities should also take the low affective commitment of academic staff very seriously. This means that academics do not have strong ties to their universities and in today's competitive labor market, they can be willing to change university or even to leave academia entirely.

**Funding:** This research was supported by the development program 'International Mobility 2019' of The Ministry of Education, Youth and Sports, received by Palacký University Olomouc, Czech Republic and by the grant IGA FF 2019 002 Shifts in Entrepreneurial Approaches in the Contemporary Economy.

**Conflicts of Interest:** The author declares no conflict of interest.

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
