# Peer review of "Does Academic Publishing Lead to Work-Related Stress or Happiness?"

_publications, doi:10.3390/publications7040066_

Round 1

Reviewer 1 Report

This is a really nice paper exploring the causes of work-related stress and happiness for academics. The context for the study is explained well, with a thorough and relevant literature review, and the method and results are clearly presented. The work is novel as far as I know - I have seen no similar studies. 

I have just a few relatively minor comments:

L33 - "..and it raises many questions" - It would be good to mention what some of these questions are I think.

Method - I think a little more detail about the coding process would be useful. Were the interviews transcribed by the researcher? Was the coding done by just one person, and if so were there any techniques used to verify the coding (e.g. asking another researcher to code some transcripts)? If more than one researcher did the coding, were any tests of inter-coder reliability undertaken?

Results - In the Conflict row of Table I assume you are setting up the tensions between different factors - so maybe "vs." (for "versus") would be clearer than "x"? (e.g. " teaching vs. publishing" instead of "teaching x publishing")

Discussion - I did wonder whether at some point in the discussion of results you could refer back to the literature you introduce in the introduction. To what extent is the stress and happiness experienced by academics different/similar to the experience of people in other professions? Is there anything unique about the academic experience with regard to happiness and stress?

Reviewer 2 Report

This is a well-written and interesting paper. I have only a few comments I would like the author to address.

Line 66. The author refers to "we found" but as there is only one author listed this should be in the singular or third person.

While I understand the need for anonymity could some more demographic information be included - i.e. age, years of tenure etc. This would be helpful in interpreting the results.

Line 348. There is a comment here that when revisions are requested by reviewers that the work is perceived as meaningless. While reviewer comments can be frustrating when they request changes which cannot be made (i.e. related to data collection methods) the intent of most reviewers, I hope, is to improve the paper by providing objective feedback. As academics we all experience the pressure to publish and do not intentionally act as a barrier to this. I would suggest that the author place some caveat around this statement so that it is not too broad. The peer-review process, with all of its flaws, is an important part of academic work.

The author has identified the limitation that the study has been conducted in only one country and therefore the results may not be generalisable. It would be appropriate for the author to propose that their study could be replicated more widely so that the results could have more general relevance outside the Czech Republic.

Reviewer 3 Report

“Does Academic Publishing Lead to Work-related Stress or Happiness?”

I appreciate this paper; and the comments by “Lucia” on pp. 6–7 make me all the more want to lobby for its acceptance and publication. However, I have a few broader and some more specific recommendations that the author may take into consideration while revising the piece. Ultimately, I’d like to see a slightly finer focus within the piece. Perhaps just a few adjustments to the text and its presentation are all that are truly necessary—most notably within the method section—if the goal is to present the results of research instead of a more anecdotal essay on the topic.

Broadly, three large bodies of literature exist that could shed some light on the issues addressed in this paper: (1) the large literature on writing for scholarly publication, which frequently describes the stresses and challenges of the writing act; (2) the literature on academic work-life satisfaction, which often addresses the role of writing for publication; and (3) the (growing) literature on writing in English as a secondary language for academic publication, which seems especially germane to the case at hand.

A seminal work in the first category (writing for scholarly publication) is Robert Boice’s Professors as Writers: A Self-Help Guide to Productive Writing (New Forums Press, 1990), which presents more research than the subtitle might suggest. Scores of additional books have been published over the past three decades, but many are focused on the individual experiences (of the authors). Three of the most useful titles for the purposes of this piece might be Wendy Laura Belcher’s Writing Your Journal Article in Twelve Weeks: A Guide to Academic Publishing Success, 2nd ed. (University of Chicago Press, 2019); Joli Jensen’s Write No Matter What: Advice for Academics (University of Chicago Press, 2017); and Helen Sword, Air & Light & Time & Space: How Successful Academic Write (Harvard University Press, 2017). Imad Moosa’s Publish or Perish: Perceived Benefits versus Unintended Consequences (Edward Elgar, 2018), could provide additional context about the pressures to publish. Less useful, though definitely in the same vein as this piece, is Dannelle Stevens’s Write More, Publish More, Stress Less! Five Key Principles for a Creative and Sustainable Scholarly Practice (Stylus, 2019). And I have not yet seen a copy of Barbara Sarnecka’s self-published Writing Workshop: Write More, Write Better, Be Happier in Academia (2019). Finally, freely available in the widely read (in North America) Chronicle of Higher Education is Michael Dooley and Kate Sweeny’s “advice” piece from September 12, 2017, titled “The Stress of Academic Publishing” (https://www.chronicle.com/article/The-Stress-of-Academic/241156). As these last titles indicate, others are clearly thinking about the intersection of writing, stress, and happiness within the lives of academics. I’ve not even touched on the scholarly journal articles addressing the topic (simply because I elect to engage with material that’s more widely accessible).

Scholars of (or commentators on) higher education have been writing about academic work-life satisfaction (the second category) for decades—and the role of publishing is often invoked therein. See the 1939 Report on Some Problems of Personnel in the Faculty of Arts and Sciences, by a special committee appointed by the president of Harvard University (the first instance of an acknowledgment of “publish or perish,” as far as I am aware); Logan Wilson’s genre-setting Academic Man: A Study in the Sociology of a Profession (Oxford University Press, 1942); Tony Becher and Paul Trowler’s landmark Academic Tribes and Territories: Intellectual Enquiry and the Culture of Disciplines (Open University Press, 1989); William Tierney’s Assessing Academic Climates and Cultures (Jossey-Bass, 1990); and, much more recently, Alexander Clark and Bailey Sousa’s How to Be a Happy Academic (SAGE, 2018); and Kate Woodthorpe’s Survive and Thrive in Academia: The New Academic’s Pocket Mentor (Routledge, 2018).

Finally, the literature on writing in English as a secondary language recently seems to have multiplied. An accessible place to start might be Ciaran Sugrue and Sefika Mertkan’s Publishing and the Academic World: Passion, Purpose and Possible Futures (Routledge, 2016). Three recent books that were on my desk just yesterday—their bibliographies would be gold mines—are James Corcoran, Karen Englander, and Laura-Mihaela Muresan’s Pedagogies and Policies for Publishing Research in English: Local Initiatives Supporting International Scholars (Routledge, 2019); Karen Englander and James Corcoran’s English for Research Publication Purposes: Critical Plurilingual Pedagogies (Routledge, 2019); and Elena Sheldon’s Knowledge Construction in Academia: A Challenge for Multilingual Scholars (Peter Lang, 2018). I am not an expert on the corpus of writing that aligns with linguistics or the teaching of English as a second language; but that body of work is vast (and some, I must admit, is rather impenetrable to the non-specialist), and I know it touches upon many of the points brought up in this piece.

Anyway, I’m pointing out this literature because it suggests three elements that I feel could use more illumination or interrogation in the piece at hand: (1) the respondents’ orientations to writing (in any language: some people love to write, and others don’t); (2) the respondents’ academic disciplines (and the role of writing therein); and (3) the respondents’ orientations to writing in English. I will mention these ideas again later—but more clarity on the respondents as embodied academics would help me situate their narratives and the analysis thereof.

Now I will share some comments that came to mind as I read the manuscript. The points that I do feel should be addressed in revision will be marked as “IMPORTANT.”

The introduction feels rather cursory to me. I suppose I’m reacting to the use of rather generic sources—since I know that writing specifically about stress within the academic workplace exists. I am not expecting a rewrite: I am just pointing out that better-suited literature exists.

Line 43: Do you mean “ability” instead of “inability” here? (If there were discrepancies between demands and inabilities, I can’t quite imagine how such a situation would play out. Logically, I’m stumped.)

Line 66: This “we” threw me off. How many researchers were involved in the study? What were their roles?

In the method section, the passive voice seems unduly restrictive. Who conducted the research? And I’m left with many questions—and several of these are, in fact, IMPORTANT and should be addressed, if the goal is to be presenting the results of a research study instead of a thought piece supported by anecdotal evidence: In what language did these “conversations in everyday life” transpire? How long did they last, on average? Were they recorded and then transcribed (and then translated)? (If so, by whom?) Where did they take place? In a public space or behind closed doors? Were they all conducted face to face, or were some via phone or otherwise mediated? Did the same researcher direct all of the conversations? How were the participants selected or recruited? To what extent was the purpose of the research revealed to the participants prior to the conversations? What fields do the participants represent? Did any of the participants study outside the Czech Republic (or, more to the point, had any studied in English-speaking countries)? What was the average number of prior English-language publications for the participants? Was the coding triangulated or cross-checked at any stage?

I remain slightly unclear as to whether publishing in Czech would have been acceptable to the participants. The question of language, in my view, certainly complicates the matter of stress and is one that should be interrogated a bit more. Again, some people like to write; others don’t. And some disciplines require more writing (or greater proclivity for writing) than others. If you compound the necessity (perceived or real) of publishing in English-language venues, I can certainly understand how individuals could become anxious upon thinking of the prospects.

Line 195: Change “associated” to “associate” here.

In the transcripts, note the inconsistency of the placement of parenthetical categories vis-à-vis closing punctuation marks. I could quibble with a number of the categorical assignments—but that, I suppose, is one of the benefits of transparency with the data and your coding arrangements. (Translation: By presenting the data and the assigned codes, you’re effectively asking for readers to argue over some of the fine points.)

“Lucia” mentions an argument over the definite article; I’ll point out only two that I would remove: line 11 (“of the workers”) and line 397 (“to leave the academia entirely”). Otherwise, I am most impressed with the text itself. (For the purposes at hand, I am perfectly satisfied with functional English over elegant English.)

In your discussion of the limitations (line 371), you point out that all of the participants are from one country. Indeed! Why not revise the title of this article to address that fact? (Every time I make this sort of suggestion, I seem to get pushback. Perhaps I am simply in a privileged position—in the United States—to be able to suggest more specificity? If not the title, then, at least the abstract or keywords could be modified to indicate the context. My view, in short, is that one should be proud of one’s origins. In no way is the contribution of this piece lessened by greater specificity of the cultural context in which the research was conducted. In fact, I would argue that the specificity in the metadata would make the research more attractive to researchers seeking to expand the scope of their understandings and cultural perspectives.)

Now, as you can see, my most substantive comments focus on the method of the study presented in this piece. In the absence of knowing additional contextual matters about the participants and the nature of the “conversations,” I am not particularly positioned to comment on the analysis or discussion: my guesses would be as good as anyone’s here. To that end, then, I trust the researcher’s analyses: who am I, anyway, to presume knowledge of the inner workings of the higher education system in the Czech Republic, particularly as such situations would affect the outlooks and wellbeing of faculty members employed therein? Because I know that so many personal factors contribute to individuals’ perceptions of writing-related stress in the United States, though, I would imagine that just as many complications would afflict the lives of academics in the Czech Republic, and I would be reluctant to jump to conclusions without additional contextual material. Self-perception and orientation and attitudes toward writing inform many of the diagnoses in the self-help literature written for Anglophone audiences. In other words, scholars who think they are “skilled” writers are more likely to “enjoy” the process of writing for publication, regardless of the requirements or expectations. Moreover, scholars who accept that their external identities are largely shaped through their published work realize the import of devoting time and attention to the craft of writing for publication. Even the most prolific authors would likely not say that anything about their writing is necessarily “easy”: writing is in fact work—even writing in one’s native language. Happiness—at least according to Clark and Sousa (2018)—is likely to be felt by academics who foster excellence as a habit, stay organized, manage their time wisely, set ambitious yet reasonable goals, and work strategically to maximize productivity within the confines of their working environments. These orientations and approaches greatly affect writing for publication (as well as teaching, service, and other engagements). In short, the mechanisms that contribute to happiness (or stress, for that matter) are notably complex—another reason why greater focus in this piece on the context of writing for publication (in English) within the Czech Republic would help to ground and substantiate the findings.
